# A Deep Learning Approach to Downscale Geostationary Satellite Imagery for Decision Support in High Impact Wildfires

Nicholas F. McCarthy [†] , Ali Tohidi [‡] , Yawar Aziz [§] , Matt Dennie, Mario Miguel Valero *,[‡] and Nicole Hu *

One Concern, Inc., 855 Oak Grove Ave, Menlo Park, CA 94025, USA; nick.mccarthy@cfa.vic.gov.au (N.F.M.); ali.tohidi@sjsu.edu (A.T.); yawarz@google.com (Y.A.); mdennie@oneconcern.com (M.D.)
* Correspondence: mm.valero@sjsu.edu (M.M.V.); nicole@oneconcern.com (N.H.)
† Current address: Country Fire Authority, 8 Lakeside Drive Burwood East, Melbourne, VIC 3151, Australia.
‡ Current address: San Jose State University, 1 Washington Sq, San Jose, CA 95192, USA.
§ Current address: Google LLC, 1600 Amphitheatre Parkway, Mountain View, CA 94043, USA.

**Abstract:** Scarcity in wildland fire progression data as well as considerable uncertainties in forecasts demand improved methods to monitor fire spread in real time. However, there exists at present no scalable solution to acquire consistent information about active forest fires that is both spatially and temporally explicit. To overcome this limitation, we propose a statistical downscaling scheme based on deep learning that leverages multi-source Remote Sensing (RS) data. Our system relies on a U-Net Convolutional Neural Network (CNN) to downscale Geostationary (GEO) satellite multispectral imagery and continuously monitor active fire progression with a spatial resolution similar to Low Earth Orbit (LEO) sensors. In order to achieve this, the model trains on LEO RS products, land use information, vegetation properties, and terrain data. The practical implementation has been optimized to use cloud compute clusters, software containers and multi-step parallel pipelines in order to facilitate real time operational deployment. The performance of the model was validated in five wildfires selected from among the most destructive that occurred in California in 2017 and 2018. These results demonstrate the effectiveness of the proposed methodology in monitoring fire progression with high spatiotemporal resolution, which can be instrumental for decision support during the first hours of wildfires that may quickly become large and dangerous. Additionally, the proposed methodology can be leveraged to collect detailed quantitative data about real-scale wildfire behaviour, thus supporting the development and validation of fire spread models.

**Keywords:** decision support; fire progression; machine learning; remote sensing; wildland fire

## 1. Introduction

In recent years, several regions throughout the world have suffered from the devastating consequences of wildfires [1]. While very large fires are generally infrequent, these rare events account for the majority of annually burned areas and can pose threats to human settlements located at the Wildland–Urban Interface (WUI). This is compounded by the concept known as the "wildfire paradox" wherein wildfire suppression policy originally intended to reduce the impacts of wildfire on communities leads to the accumulation of available fuel, which increases the probability of large fires occurring with higher intensities than without the suppression policy [2]. Strategic decision support in fast-moving, destructive WUI fires is notably different from other "routine" wildfires due to the type of emergency response they require. Typical wildland firefighting is characterized by well-organized modules assigned to objectives and divided by function to control, slow, or monitor the fire. On the contrary, fires such as the Northern California fires of 2017 (North Bay) and 2018 (Camp Fire) require evacuations and structure defense with complex and competing dispatch requirements to be attended by multiple agencies [3]. Additional challenges include the delays (at least one day) in establishing an incident command organization capable of dealing with the complexity and size of the incident, and the ability of

aerial reconnaissance and suppression to operate, given extreme wind conditions and time of day.

In this context, there is a great need for decision support tools that inform decision making during wildfire emergencies. Specifically, short-term predictions of the fire perimeter location, the rate of spread, and the fire intensity are crucial information. However, quantitative information about the fire state—location, velocity, and intensity—is generally not available in real-time during a wildfire event [3]. Even offline time-resolved validation data-sets are scarce, and the difficulties with collecting such data are well-documented [4,5]. This severe data void results in a fundamental lack of knowledge and intelligence to support quantitative fire behavior characterization and decision making. In this regard, even fire models are not able to accurately estimate the state of behavior in a general scenario; see [6–8] for previous analyses of current fire modeling capacities. This is mainly due to the complex and multi-scale dependencies in wildfire propagation mechanisms which make conventional modeling practices susceptible to compounding uncertainties [7,9]. The required fuel, weather, and terrain properties for modeling fire spread are difficult to measure experimentally and, often extremely challenging to accurately estimate in real wildfire scenarios. Furthermore, one of the main sources of uncertainty in the forecasts obtained by the operational fire models is the lack of knowledge about initial ignition point(s) which consequently results in significant under or over predictions [10]. The necessity to reduce uncertainties in the forecasts has derived the development of data-driven models [11–16]; however, the persistent lack of observational data with adequate spatial and temporal resolutions limits their functionality to only experimental settings.

Remote sensing provides invaluable opportunities to gather fire behavior intelligence, mostly in the form of fire location, rate of spread, and radiated energy [17]. In this regard, the Earth Observation (EO) satellites are highly reliable and provide a consistent stream of data. EO sensors have been thoroughly validated and their behavior can be predicted. The main limitation of space-borne wildfire monitoring sensors is that they lack the required spatial or temporal resolution or both.

The most widely used space-borne EO platforms work on either Geosynchronous Equatorial Orbits (GEO) or low-Earth, polar sun-synchronous orbits (which we refer to as Low Earth Orbits—LEO—for simplicity). On the one hand, GEO sensors allow observing the Earth at high temporal resolution but they have significant restrictions on spatial resolution. Moreover, wildfire monitoring sensors usually operate in mid-wave and long-wave infrared ranges, which further limits image spatial resolution. On the other hand, LEO satellites orbit closer to the Earth but they typically provide only 2–4 snapshots of the same area per day [18,19]. Both GEO and LEO platforms have been used operationally for wildfire applications, but their use requires a compromise between pixel size and temporal resolution [20–26]. This trade-off has so far limited the use of space-borne remote sensing imagery in wildfire management to detection and long term monitoring, as opposed to sub-daily real-time mapping [27].

Hence, this paper provides a scalable solution for collecting near-real-time high-resolution data of the wildfire progression. A deep learning framework—extending the work previously presented by the authors in [28]—that use LEO and GEO satellite multi-spectral imagery, land use, vegetation, and terrain data is proposed. Our model is able to estimate the evolution of an active fire perimeter with 375-m resolution at 5-min intervals. The rest of this paper is structured as follows: Section 2 details the designed data structure, the deep model that lies at its core and its training process, as well as the required post-processing tasks. Afterwards, Section 3 describes the operational implementation of our solution. Finally, Section 4 provides validation results, followed by a discussion of their implications in Section 5.

## 2. Methodology

Our methodology is conceptually based on a statistical downscaling algorithm used to infer high spatial resolution probabilities from GEO satellite imagery. The nominal output

pixel size is 375 m. At this resolution, each raster pixel is assigned a probability of being burnt at a certain time based on the latest available satellite imagery as well as vegetation, land use and terrain information of the area. Subsequently, pixel probability distributions are thresholded to compute the most probable location of the active fire area at a given time. Finally, time series analysis of fire location allows tracking the fire perimeter evolution and measuring its rate of spread.

In order to achieve this goal, GEO satellite imagery is retrieved for the area of interest with a temporal resolution of 5 min. Six different multispectral bands are used, ranging from visible red (0.64 μm) to long-wave infrared (12.3 μm). The spatial resolution of this imagery lies between 0.5 km and 2 km. GEO data is combined with spatially-explicit information about land use, terrain and vegetation and input into a deep learning model for fire segmentation. Input data sources are described in detail in Section 2.1.

The deep learning segmentation model is trained on LEO remote sensing data before operation. Fire detection products provided by current LEO platforms constitute a valuable source of training data because they provide pixel-wise fire information that can be used for labelling purposes. While LEO sensors lack temporal resolution, they provide significantly higher spatial resolution than GEO platforms. Furthermore, the fact of using remote sensing products for unsupervised labelling allows automation of the training process, which facilitates continuous improvement of the segmentation algorithm and seamless integration of newly available data. Figure 1 displays a schematic overview of the complete system structure.

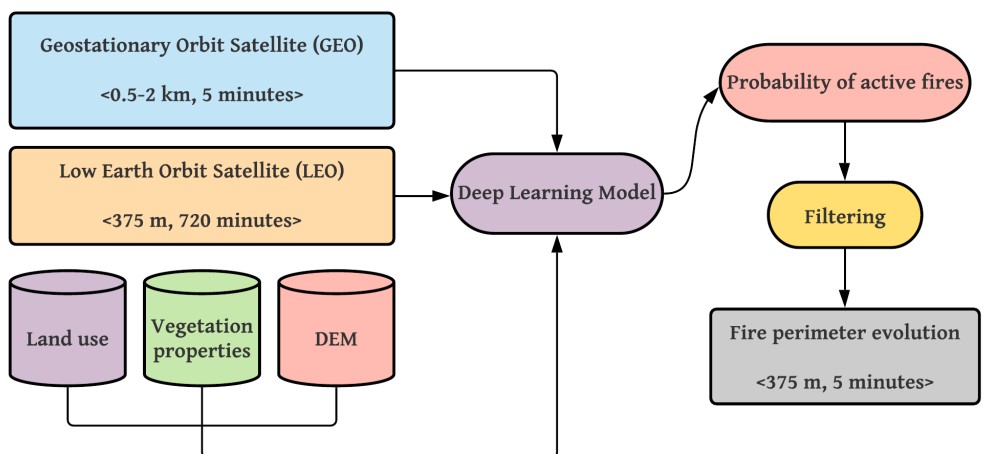

**Figure 1.** Block diagram of the statistical downscaling framework used for high-resolution continuous monitoring of active wildland fires.

The deep learning model consists of a U-Net Convolutional Neural Network (CNN) whose design and training process are described in Sections 2.2 and 2.4, respectively. The features used by the model are detailed in Section 2.3, and Section 2.5 describes model outputs and the required post-processing steps.

### 2.1. Input Data

Input features used by our fire monitoring algorithm can be grouped into quasi-static and dynamic. Quasi-static features include terrain, vegetation and land use information, whereas dynamic features are provided by GEO satellite imagery. Terrain, vegetation and land use data were manually downloaded from the LANDFIRE LF.2.0.0 data repository [29]. Feature layers used for model training and operation include fuel model [30], elevation (DEM), slope (SLP), canopy cover (CC), and canopy height (CH). Because this information is not expected to vary frequently in time, our system considers these layers static. Nonetheless, stored data can be easily updated when necessary.

Dynamic fire classification features are derived from GEO multispectral imagery provided by the National Oceanic and Atmospheric Administration (NOAA) Geostationary Operational Environmental Satellites (GOES). The primary sensor used for this purpose is the Advanced Baseline Imager (ABI). Installed aboard GOES-16, ABI works in 16 different spectral bands with spatial resolutions ranging from 0.5 km to 2 km. GOES-16 provides a full disk image every 10 min, a CONUS image every 5 min and images from mesoscale domains every 60 s. Our fire monitoring system ingests ABI CONUS imagery every 5 min. Spectral bands employed are detailed in Table 1. Data from these bands is retrieved through the ABI-L2-MCMIPC product [31] and ingested automatically during the monitoring workflow.

**Table 1.** Advanced Baseline Imager (ABI) channels used for dynamic feature generation. Specifications reproduced from [26].

| ABI Band | Central Wavelength (μm) | Type | Nickname | Best Spatial Resolution (km) |
|---|---|---|---|---|
| 2 | 0.64 | Visible | Red | 0.5 |
| 5 | 1.6 | Near-infrared | Snow/ice | 1 |
| 6 | 2.2 | Near-infrared | Cloud particle size | 2 |
| 7 | 3.9 | Infrared | Shortwave window | 2 |
| 14 | 11.2 | Infrared | Longwave window | 2 |
| 15 | 12.3 | Infrared | "Dirty" longwave window | 2 |

LEO labelling data necessary for training is obtained from the Visible Infrared Imaging Radiometer Suite (VIIRS) 375 m active fire detection data product (VNP14IMGTDL_NRT) [32,33]. The VIIRS sensor orbits aboard the joint NASA/NOAA Suomi National Polar-orbiting Partnership (Suomi-NPP) satellite. Suomi-NPP is a LEO satellite that observes the Earth's surface twice every 24 h. The active fire product belongs to a group of Near-Real Time (NRT) data products which become available shortly after their acquisition. Additionally, NRT data is further processed and archived in the form of higher level products. VIIRS fire detection data is provided as sparse arrays that include pixel location, time of fire detection, confidence on fire detection and an estimation of Fire Radiative Power (FRP), among others. Temporal and spatial selections of this sparse array are read by the system as needed during model training.

*2.2. Deep Learning Model Architecture*

U-Net is a CNN architecture designed for image segmentation. It was originally proposed by Ronneberger et al. [34] and it quickly became one of the reference algorithms most frequently used today for that purpose. The U-Net has outperformed its predecessors in several remote sensing applications (e.g., [35–37]). Its name was derived from the U-shape frequently used to represent its architecture.

A U-Net is generally composed of a contracting path (feature detection) followed by a usually symmetric expansive path (feature localization) [34]. The expansive path is implemented through upsampling operators followed by convolution with the original, high-resolution feature layers from the contracting path. An important property of the U-Net is the fact that it does not have any fully connected layers, which allows splitting the input image into tiles that overlap with each other. This overlap-tile strategy allows the seamless segmentation of arbitrarily large images and avoids resolution limitations caused by GPU memory [38]. Our U-Net implementation consists of a total of 27 layers that combine convolution, non-linear activation and pooling operations. The order in which they are applied is displayed in Figure 2 together with the number of channels used in each step. The Python [39] deep learning API Keras [40] was used to configure this architecture.

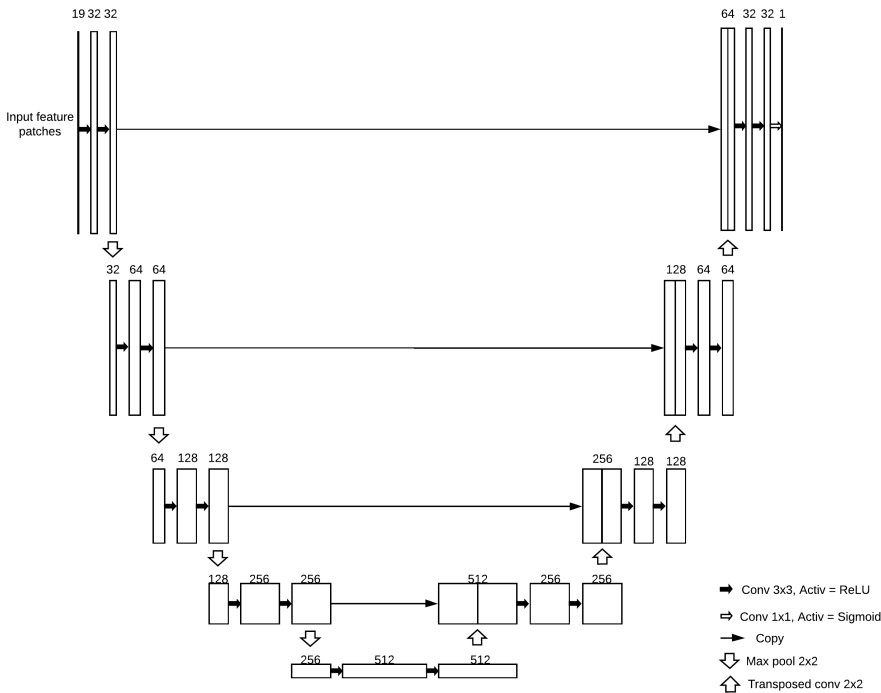

**Figure 2.** U-Net architecture used for fire segmentation.

### 2.3. Pre-Processing and Feature Engineering

The designed U-Net operates on a set of features derived from the input data described in Section 2.1. Dynamic features include intensity retrieved from spectral channels 2, 5, 6, 7, 14 and 15 in the GOES-16 ABI-L2-MCMIPC product. Additionally, GOES-16 ABI data is augmented through combinations of the original bands. Feature layers added to the original data include channel divisions (6/5, 7/5, 7/6, 14/7), channel differences (7-14, 6-5) and Z-score computed for every channel independently. Furthermore, the time of the day at which the imagery was acquired is added as an extra feature in order to account for changes in lighting conditions and sunglint distortions. The above modifications yield a total of 19 dynamic feature maps that are created every 5 min and input into the ML fire classification module.

Four additional static features reinforce automated classification by including information about land use, terrain and vegetation. Specifically, terrain is characterized through elevation and slope, whereas canopy height is used to account for vegetation. Other features such as canopy cover and vegetation type are expected to be added in the future. All feature maps are resampled to the target output resolution before being input to the model.

### 2.4. Model Training

U-Net training is performed using the Adam optimizer [41] with a learning rate of 0.001 and the beta values proposed in [34]. A custom loss function is employed, defined as a cross-entropy weighted according to the label weights. Pre-processed feature maps are split into patches of size $50 \times 50$ pixels. Metrics evaluated during model training and testing include accuracy, F-measure and the Area Under the ROC Curve (AUC). Fifty epochs are used to train the model in batches of size 30. The system is prepared for periodic training to incorporate new available data. Label data is automatically downloaded from public repositories and ingested into the system when made available. Re-training can be scheduled either periodically or every time new data is added to the fire detection repository. Currently, active fire detection information is downloaded from the NASA Near Real-Time Data and Imagery service, Fire Information for Resource Management System (FIRMS).

*2.5. Post-Processing*

Active fire probability fields provided by the deep learning model are subject to several post-processing steps in order to estimate fire perimeter evolution. Firstly, low-pass spatial filtering is applied through convolution with a median mask of size 5. Median filtering is applied to each output frame individually. Subsequently, consecutive frames are stacked in 30-min windows for time series analysis. Temporal noise removal is accomplished through a Savitzky–Golay filter [42] of second order. Filtered probability 2D distributions are then thresholded to classify pixels into active fire and background. A sensitivity analysis was conducted to determine the optimum probability threshold. The optimum thresholding value suggested by such sensitivity analysis was 0.7.

Active fire pixels detected at each time step are added to the fire footprint resulting from the previous time step. Finally, fire contour at each time step is computed and exported as a vector geospatial layer for subsequent analysis.

## 3. Operational Implementation

The practical implementation of the described system has been optimized for operational deployment. It is based on Cloud Service Providers (CSPs) and it leverages the advantages of cloud compute clusters, software containers, and multi-step pipelines optimized for parallel workflow computing. We leveraged a CSP on Google Cloud, orchestrated by Kuberentes [43], to build a compute cluster which consists of a pool of nodes (virtual machines) networked together in a fashion to allow compute workloads to be distributed across the cluster. A workflow was created using Argo [44] to manage task scheduling and mapping of the model and to train the machine learning model every two weeks. Because the workflow ingests the latest available data each time, the model can improve automatically through this process. Another workflow was created to run the model for real time monitoring on a five minute schedule. Finally, a third workflow was created to validate the model's accuracy.

The machine learning pipeline was designed to output data which conforms to well-known geospatial file formats to maximize interoperability. Output file formats used are RFC 7946 (GeoJSON) and Cloud Optimized GeoTIFF [45].

## 4. Validation

The proposed monitoring system was validated in five wildfires selected among the most destructive occurred in California in 2017 and 2018 (Table 2). The methodology described in Sections 2 and 3 was used to reconstruct fire progression at 30-min intervals and 375-m resolution.

**Table 2.** Wildfires used for validation of the proposed methodology.

| Event Name | Dates | Location | Burned Area (km²) | Deaths | Number of Structures Destroyed |
|---|---|---|---|---|---|
| Atlas | 8–28 October 2017 | Napa County (CA, USA) | 207 | 6 | 781 |
| Camp | 8–25 November 2018 | Butte County (CA, USA) | 620 | 85 | 18,804 |
| County | 30 June–17 July 2018 | Yolo and Napa Counties (CA, USA) | 365 | 0 | 20 |
| Delta | 5 September–7 October 2018 | Shasta and Trinity Counties (CA, USA) | 256 | 0 | 20 |
| Tubbs | 8–31 October 2017 | Napa, Sonoma, and Lake counties (CA, USA) | 149 | 22 | 5643 |

The estimated fire isochrones were validated against the fire perimeters reported by the Geospatial Multi-Agency Coordination (GeoMAC) [46]. The GeoMAC is a web-based application developed and maintained by the United States Geological Survey (USGS). It collects fire spread information in order to provide fire managers with improved situational awareness. Fire perimeter data is updated daily based on input from incident intelligence sources, GPS data, infrared imagery from fixed wing aircraft and satellite imagery [47]. Therefore, GeoMAC perimeters represent the best Ground Truth data about wildfire evolution available today in the US. They leverage data provided by other public agencies such as the National Interagency Fire Center (NIFC), the US Forest Service (USFS), the National Aeronautics and Space Administration (NASA), the Bureau of Land Management (BLM) and the National Oceanic and Atmospheric Administration (NOAA).

The GeoMAC database was also leveraged to define the spatio-temporal windows used for model training. Historical data was collected for over 200 wildfire incidents recorded in California in 2017 and 2018. Their dates and spatial extent were used to constrain the search for VIIRS active fire detections. Because the validation fire events indicated in Table 2 were present in the training database, the model was re-trained for every validation fire following a leave-one-out cross-validation scheme.

Figure 3 shows a qualitative visualization of the achieved results, with half-hourly estimated fire evolution superimposed on the first perimeter reported in the GeoMAC database for each fire.

Table 3 summarizes the similarity metrics computed between predicted and observed fire perimeters. Precision and Recall, respectively, measure the system's robustness in the presence of noise leading to overprediction and its ability to retrieve fire pixels avoiding underprediction. The Threat Score, similar in formulation to Jaccard's index [48] and sometimes called Critical Success Index (CSI), accounts for both false positives and false negatives and it is a reliable measure of fire perimeter similarity [13].

**Table 3.** Performance metrics computed for the fire isochrone closest in time to the available GeoMAC perimeters. TP, true positive; FP, false positive; FN, false negative. Precision = TP/(TP + FP); Recall = TP/(TP + FN); Threat Score = TP/(TP + FP + FN).

| Fire Event | Precision | Recall | Threat Score |
|------------|-----------|--------|--------------|
| Atlas | 0.58 | 0.89 | 0.54 |
| Camp | 0.73 | 0.87 | 0.66 |
| County | 0.90 | 0.70 | 0.65 |
| Delta | 0.55 | 0.87 | 0.50 |
| Tubbs | 0.49 | 0.99 | 0.49 |

Validation results shown in Figure 3 and Table 3 indicate a tendency to overpredict. In the context of classification, the higher the values of Precision, Recall, and Threat Score the better is the classifier. Interpreting the results is closely tied to the downstream applications. For emergency response management applications, higher values (closer to 1) of Recall and Threat Score are desirable which effectively implies smaller samples (pixels) that are falsely predicted negative (not on fire). In the context of fire behavior analysis, accurate estimation of the fire line is important which calls for low false positives and false negatives. As a result, the Threat Score is a more suitable metric for assessing the performance of the model for estimating the fire line progression. As shown in Table 3, Recall is generally high and Precision values drop significantly in some cases, which produces a consequent decrease in Threat score. We suspect that the most probable cause for this behaviour is false positives produced by hot gases present around the fires. The majority of these fires are wind-driven, which causes smoke plumes to tilt, usually in the main direction of fire spread.

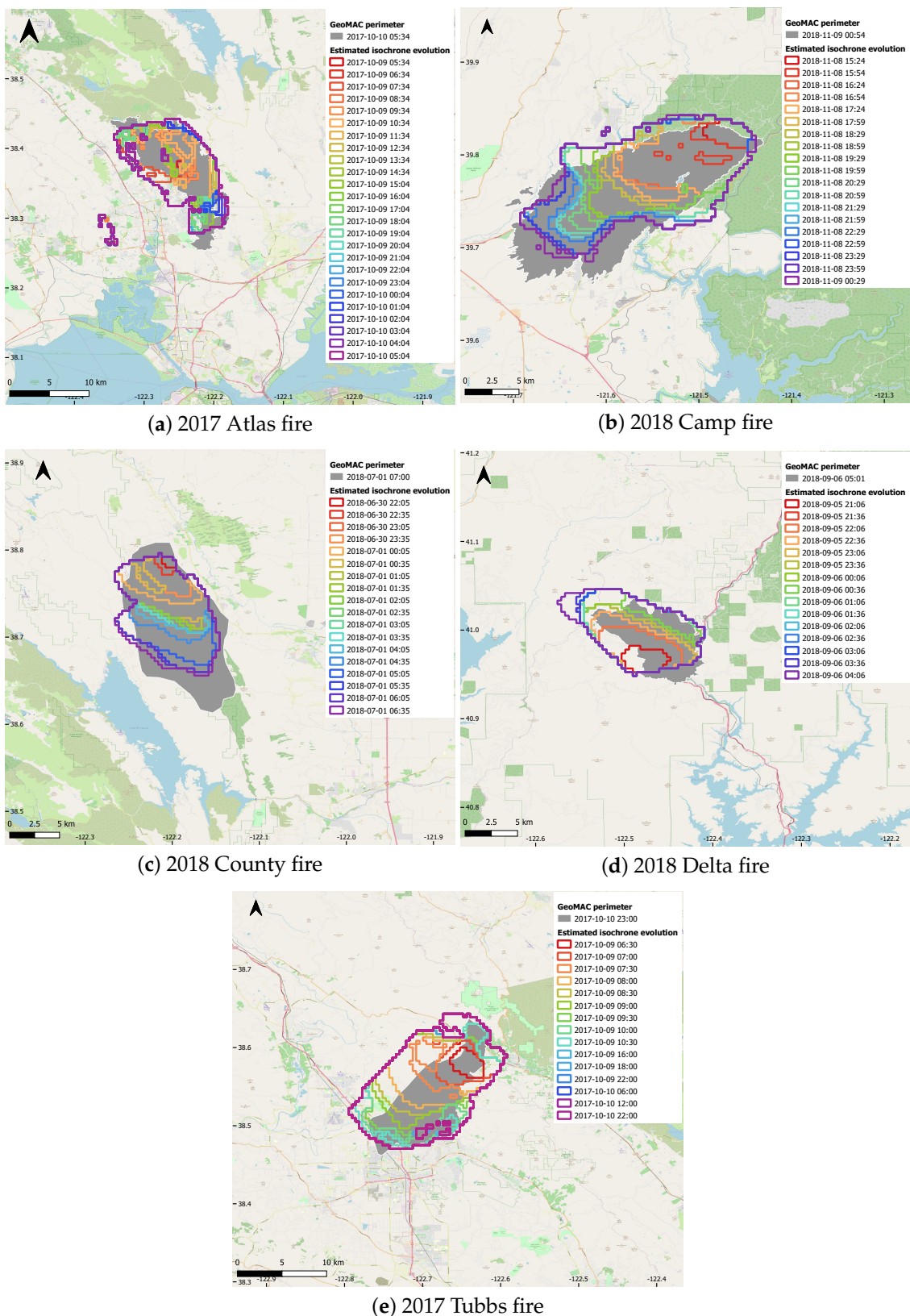

**Figure 3.** Half-hourly fire progression estimated using the proposed algorithm (color lines), compared against the first perimeter available in GeoMAC (grey polygon). Some intermediate isochrones have been hidden for visualization purposes. All times are UTC. Background map provided by OpenStreetMap [49].

In addition, Figure 4 demonstrates the applicability of our algorithm for public safety decision support. All non-medical 9-1-1 calls received by the Sonoma public-safety answering point (emergency services dispatchers) during the Tubbs fire are superimposed on the isochrones obtained by the presented methodology. The 9-1-1 call volume shown in Figure 4 represents 20 times the average volume of calls in a 24 h period for dispatchers in the county, while the main fire spread event occurred in less than 6 h [50]. This translated into complete exhaustion of first responder resources, such as fire engines and ambulances, and prolonged delays in response times to requests for help. In such a situation, "Pre-Arrival Instructions" from call takers require aiding callers to navigate to safety, which the presented methodology, despite its limitations, could have substantially aided in. The Tubbs fire case study illustrates the added context and situational awareness for triage that the proposed tool can provide to fire dispatchers and commanders.

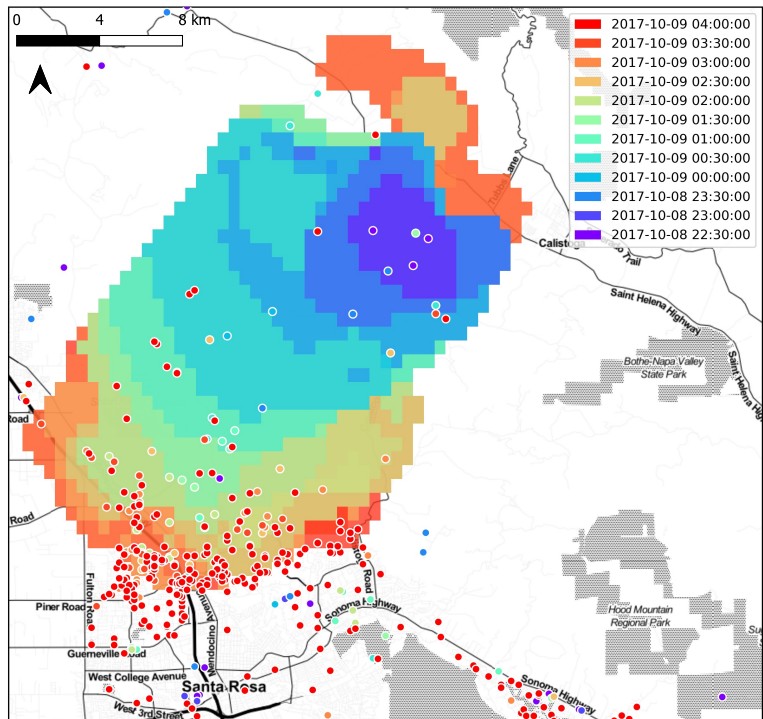

**Figure 4.** Fire progression estimated for the 2017 Tubbs fire along with all non-medical 9-1-1 calls received during the period of the fire. Fire isochrones and 9-1-1 calls are color mapped using the same time legend, noting many calls occur after the last isochrone shown. Only the first six hours of spread are displayed, and all times are given in local time (UTC-7).

## 5. Discussion, Conclusions and Future Work

This paper presents an application of deep learning segmentation techniques for continuous monitoring of active wildfires. U-Net CNNs have received significant attention in computer vision problems due to their outstanding segmentation performance, especially in large images that need to be split and processed in parallel. We extended the U-Net application to geostationary remote sensing imagery in order to improve the spatial resolution of active wildfire detections, thereby allowing, for the first time to the best of our knowledge, high-resolution active wildfire monitoring from space.

The proposed algorithm leverages complementary properties of GEO and LEO sensors, plus static features related to topography and vegetation. In this manner, fire detections are not only based on remote sensing data, but the analysis of RS imagery is further informed by physical knowledge about fire behaviour. This strategy also constitutes a novel approach in the literature, and the successful results presented here encourage further development.

Promising follow-up work includes the addition of weather variables, such as wind speed and direction, to the set of features used for segmentation.

Besides the algorithm conceptualization details, this article describes the practical implementation scheme that we designed for operational deployment. Due to the important applications of a monitoring system like this, its design has been oriented towards practical usability and runtime performance from the beginning.

The main detected limitation of the current version of the algorithm, to be addressed in the near future, is the relatively high false positive ratio, that we suspect is primarily caused by the heat signature of the convective plume. The addition of weather classification features and the expansion of the training database with time are expected to help overcome this limitation. Other planned improvements include the addition of RS imagery acquired by other modern platforms, such as GOES-17 and the Meteosat Third Generation satellites. Additionally, the use of other novel deep learning architectures or the development of a new ad-hoc model may also improve the achieved performance.

The current version of this algorithm is targeted at decision support during wildfires that unfold in less than one day and grow in thousands of acres. The lack of ability to characterize extreme behaviour during such rapidly unfolding fire events, even in simple metrics of hourly area growth, severely hinders decision making. This algorithm demonstrates that it is possible to automatically generate the intelligence required for public safety decisions in a scalable manner by using existing satellite data. Furthermore, high-resolution fire progression data provided in almost real time can be leveraged to improve the fire spread forecasts issued by operational models.

**Author Contributions:** Conceptualization, N.F.M., A.T. and N.H.; funding acquisition, N.H.; methodology, N.F.M., A.T, Y.A., M.D. and M.M.V.; project administration, N.H.; software, N.F.M., A.T., Y.A., M.D. and M.M.V.; validation, N.F.M., A.T. and M.M.V.; writing—original draft preparation, N.F.M., M.D. and M.M.V.; writing—review and editing, N.F.M., A.T. and M.M.V. All authors have read and agreed to the published version of the manuscript.

**Funding:** Financial support for this project was provided by One Concern, Inc.

**Institutional Review Board Statement:** Not applicable.

**Informed Consent Statement:** Not applicable.

**Data Availability Statement:** Not applicable.

**Acknowledgments:** 9-1-1 anonymized data was kindly provided by the Redwood Empire Dispatch Communications Authority (REDCOM). Geostationary data was provided by the open source NOAA Big Data Program, and the LEO data were provided by the NASA Earth Data program via the Fire Information for Resource Management Systems (FIRMS). The authors would also like to thank the One Concern team for their support during the development of this platform, specifically the insights provided in early stages by Seth Guikema, Abhineet Gupta, and Ahmad Wani.

**Conflicts of Interest:** The authors declare no conflict of interest.

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
