# Peer review of "A Deep Learning Approach to Downscale Geostationary Satellite Imagery for Decision Support in High Impact Wildfires"

_forests, doi:10.3390/f12030294_

Round 1

Reviewer 1 Report

The paper entitled "A deep learning approach to downscale geostationary satellite imagery for decision support in high impact wildfires" by McCarthy et al. proposes and describes an implementation of a computing infrastructure to produce improved fire growth data in (very) near real time by downscaling high-temporal/low spatial resolution imagery from GOES-R (GOES-16) to the spatial resolution of 375m VIIRS active fire detections. Alternatively one might describe the process as temporal downscaling of the very widely used VIIRS active fire data using GOES, which in its latest incarnation is much improved in its imagery interval and substantially improved in spatial resolution compared to previous geostationary earth observation satellites.  Ancillary input data is provided by vegetation cover from Landfire and terrain data from USGS 3DEP (also via Landfire). Key to the statistical downscaling is a machine learning approach using a convolutional neural network, as well as a scalable and rapidly deployable cloud architecture. 

This is an interesting topic and while the technique is only applicable to low and medium latitudes, the results look to be of a quality that might approach the fire growth prediction systems  currently in use by fire management agencies in North America. As an automated data source that only needs minimal input to start producing fire growth assessments for a new fire event (and no input from a fire behavior analyst) I would imagine that management agencies might be highly interested in test-driving this product. 

The organization of the paper is suitable, and the exposition overall clear. So while the technical and data analysis work done appears to have scientific merit and deserve publication, I have a number of rather weighty concerns that would need to be addressed before publication. 

These concerns relate to what comes across as a lack of awareness of the paradigms and practices of the applied fire remote sensing community. Some of the telling signs are harmless. For example, the authors refer to S-NPP (the satellite platform carrying one of two VIIRS sensors currently in orbit) as a low earth orbit satellite and use the LEO abbreviation throughout, while it is more common to refer to S-NPP as a polar-orbiting sun-synchronous satellite. These are in fact platforms that fulfill the LEO definition, so it is not wrong to call them that, but as far as their earth observation characteristics are concerned, it is their sun-synchronous nature as polar-orbiting platforms that makes them suitable for global observation - a quality that other LEO platforms (such as ISS for example) lack. I am not suggesting the authors get rid of the LEO label, but am putting this observation out here as a mark of awareness of the norms of the field.

Other points are more serious and really DO need addressing. In particular, as citation is concerned. The authors cite Lentille et al (2006) as a review paper on active fire remote sensing,  neglecting  Ichoku et al.'s 2012 paper "Satellite contributions to the quantitative characterization of biomass burning for climate modeling" (DOI: 10.1016/j.atmosres.2012.03.00710.1016/j.atmosres.2012.03.007), which has a newer overview. 

In section 2.1 "Input data" the only citation referring to one of the products they used is Schroeder et al. for the scientific basis of the VIIRS 375m active fire product. However the product itself is not cited even though NASA/LANCE/FIRMS make it very easy to find out how they would like to be cited and acknowledged (see https://earthdata.nasa.gov/earth-observation-data/near-real-time/citation ) . Similarly, Landfire is "cited" in a footnote (which is very much contrary to citation practices in the scientific literature - something I have not had to point out as a peer reviewer before). Landfire, like a large number of institutionally supported data providers, is quite clear about how to properly cite their data (see https://www.landfire.gov/landfire_citation.php ). The chapter also fails to list the exact product references that were used. The GOES-16 ABI data is lacking the reference for the Algorithm Theoretical Baseline Document (which is available here: https://www.star.nesdis.noaa.gov/goesr/documents/ATBDs/Baseline/ATBD_GOES-R_ABI_CMI_KPP_v3.0_July2012.pdf ) .

On a similar note, Python, Numpy, Keras etc. all provide information about how to properly cite the software for use in scientific papers. (I didn't see anything for Kubernetes, on my quick search.) In general, using a search engine of the authors' preference, "[dataset/software] how to cite in research papers" quickly will provide a dozen or so missing citations. 

The above may sound a bit like fuddy-duddy pedantry, but really isn't. Both public research agencies (NASA, NOAA, USGS in the case at hand) and open-source software development teams really *need* to be able to track use of their output in the scientific literature in order to be able to assess their impact and justify requests for funding based on the public good they provide. So please remedy the citation situation.

Last, the authors don't appear to be aware of how wildfire growth and behavior modeling is in fact implemented in operational decision support systems run by fire management agencies in North America (this reviewer is only aware of the Canadian and *some* of the US-based systems). For example in the Introduction, the authors claim "Moreover, state-of-the-art fire models are not able to accurately reproduce – let alone predict – wildfire behaviour in a general

scenario." Are the authors aware of existing operational methodologies to assess real-time fire behavior (a starting point might be https://www.firelab.org/project/behaveplus as well as works like Tymstra, C., & Bryce, R. (2010). Development and structure of Prometheus: The Canadian Wildland Fire Growth Simulation Model (Information Report NOR-X-417). Northern Forestry Centre.) ? In future work it would certainly be interesting to compare their output to fire behavior modeling output. On first glance, the results look comparable in quality. (I am aware that the authors' approach is actual observation, not just modeling.) I do not see a reason to suspect that the practical logistical limitations of an emergent catastrophical fire (getting the fire map done, but also getting the output in real time in front of the eyes of incident command) would be any less constricting for the system the authors propose than for what is in use currently. 

Therefore, while I recommend the article for publication in principle, I think that substantial additional work is necessary to make it useful in the literature of its field. Maybe the authors could consider adding a fire remote sensing expert to the author list, who would be aware of the deficiencies described.

Some minor line-by-line remarks :

ll 47-49: "Specifically, real-time information as well as short-term predictions of the fire perimeter location, its rate of spread, and intensity are key to allocate firefighting resources and issue evacuation orders in a timely manner." The Introduction in general is too long and goes into a number of fire facts that while dramatic contribute little to the argument. I suggest the authors start from this sentence and then add only what is necessary to illustrate this point as well as how their work contributes to it, cutting the introduction in half . 

ll 147-148: "Feature layers used for model training and operation include land use, elevation, slope and canopy height" Please provide the exact reference labels for the Landfire layers you used.

ll 148-149: "Because this information is not expected to vary frequently, our system considers these layers static." And also because vegetation and terrain data is not in fact available with an update frequency on a scale of minutes (or even days). 

ll 178ff: For the benefit of readers from the fire science and remote sensing communities the authors should consider adding references that provide accessible entrance points to the claims about properties of the U-Net architecture ("... features detected during the former can be localized during the latter...") . I note that in the preceding paragraph, the note about the performance of the U-Net also lacks a reference. 

ll 236-246: This paragraph reads somewhat like an advertisement for AWS and Google Cloud. The considerations about the expense structures of scientific computing services contribute little to the argument and should be reduced to one single sentence. Similarly , the next paragraph describing container orchestration with Kybernetes is interesting as the implementation DOEs contribute to the scalability and deployability features of the system. Maybe the passage could be largely replaced by a table listing the architecture components. 

Fig. 3: Please add a north arrow and lat/lon indicators on the axes.

Fig. 4: Please add a scale,  lat/lon indicators on the axes, and a north arrow. 

Reviewer 2 Report

I suggest just a few minor modifications to this well-presented and interesting study.

The Abstract mentions the term Deep Learning (DL) but DL is never mention in the text. Remove here or add to text. If added to the text, it needs to be described or cited.

The Abstract should include a sentence on the results. It is really just a justification and review of methods. Revise Abstract to include why, how, results, outcomes/relevance.

The Introduction is longer and more detailed then needed. I suspect it can be cut by about 1/3 and still present the ideas. Since this is a forestry journal, concepts related to fire and needs for this type of research do not need such explicit mention, and can alternatively be cited. (basically, it seems to drag on-and-on, and is not required to be so long). I will leave it to the authors to trim, if the Editors agree with me.

  • Line 96-97: This is an awkward sentence that needs revision: "Such mapping at or better than an hourly scale ..." works better (I had to re-read it several times to understand what "less" meant".
  • Line 108: Please more directly cite that this is an extension of your work: "This paper extends work previously presented by the authors (30)..."
  • Line 138: spell out CNN the first time used. The instance in the Abstract does not count. The abstract stands alone. 
  • Line 147: The link to Landfire should be included as a reference

In the paragraph starting on Line 308, please include more specific information on interpretation of the metrics in Table 3. In other words, add in what it means that the "Threat Score" is around the center of a 0 to 1 index. What does 0 mean? What does 1 mean? I have no idea if a high or low number id good or bad. Same with the other metrics.

The text starting on line 352 is new information, and should not be included in the final part of the paper (conclusions). Move it to the end of the previous section, or make a proper "discussion" and "conclusion" sections. New material never should be put at the very end.

I also am confused by the statement on line 352: Why not applicable to all wildfires? Rapidly-moving fires in areas outside the WUI can also be destructive, and require management. This technique could have broad use.

Reviewer 3 Report

This is a straightforward presentation on an important technique and topic. The authors should be commended for the quality of this research presented in this paper. I believe this paper will be very important for the wildfire research community and the authors chose appropriate extreme cases to justify their method.  

Minor Comments and Suggestions:

Figure 2: Font size could be larger. Hard to see on computer. 

Line 319: "Due to their high temperature..." I don't think plumes are high temperature. Entrainment limits the heating of the plume to lowest altitudes. Maybe rephrase? I don't think this is a problem, but maybe reconsider the wording or add caution to statement. 

Line 366: "In such a situation..." 

Please discuss in conclusions section how this technique could be implemented in a coupled fire-atmosphere modeling framework such as WRF-SFIRE. 

Also, please describe in more detail why Fig 3b (Camp Fire) perimeter is not well represented by the method. I may have missed this, but the two perimeters are not well aligned and this should be an easy case. Grass fuels, wind-driven. Please add some discussion on the differences in forecasted and observed final perimeters. 
